# Characterization of Seminal Microbiome of Infertile Idiopathic Patients Using Third-Generation Sequencing Platform

**DOI:** 10.3390/ijms24097867

**Published:** 2023-04-26

**Authors:** Sergio Garcia-Segura, Javier del Rey, Laia Closa, Iris Garcia-Martínez, Carlos Hobeich, Ana Belén Castel, Francisco Vidal, Jordi Benet, Maria Oliver-Bonet

**Affiliations:** 1Unit of Cell Biology and Medical Genetics, Department of Cell Biology, Physiology and Immunology, Universitat Autònoma de Barcelona (UAB), 08193 Bellaterra, Spain; sergio.garcia.segura@uab.cat (S.G.-S.); maria.oliver@uab.cat (M.O.-B.); 2Histocompatibility and Immunogenetics Laboratory, Banc de Sang i Teixits (BST), 08005 Barcelona, Spain; lclosa@bst.cat; 3Medicina Transfusional, Vall d’Hebron Institut de Recerca (VHIR), Universitat Autònoma de Barcelona (UAB), 08035 Barcelona, Spain; 4Grup de Coagulopaties Congènites, Banc de Sang i Teixits (BST), 08005 Barcelona, Spain; 5Instituto de Fertilidad, C. Calçat 6, 07011 Palma de Mallorca, Spain; 6CIBER de Enfermedades Cardiovasculares (CIBERCV), 28029 Madrid, Spain

**Keywords:** seminal microbiota, MinION, nanopore sequencing, Illumina, male fertility

## Abstract

Since the first description of a commensal seminal microbiome using sequencing, less than a decade ago, interest in the composition of this microbiome and its relationship with fertility has been growing. Articles using next-generation sequencing techniques agree on the identification of the most abundant bacterial phyla. However, at the genus level, there is still no consensus on which bacteria are most abundant in human seminal plasma. This discrepancy may be due to methodological variability such as sample collection, bacterial DNA extraction methodology, which hypervariable regions of *16S rRNA* gene have been amplified, or bioinformatic analysis. In the present work, seminal microbiota of 14 control samples and 42 samples of idiopathic infertile patients were characterized based on full-length sequencing of the *16S rRNA* gene using MinION platform from Oxford Nanopore. These same samples had been analyzed previously using Illumina’s MiSeq sequencing platform. Comparison between the results obtained with the two platforms has been used to analyze the impact of sequencing method on the study of the seminal microbiome’s composition. Seminal microbiota observed with MinION were mainly composed of the phyla *Firmicutes*, *Proteobacteria*, *Bacteroidetes* and *Actinobacteria*, with the most abundant genera being *Peptoniphilus*, *Finegoldia*, *Staphylococcus*, *Anaerococcus*, *Campylobacter*, *Prevotella*, *Streptococcus*, *Lactobacillus*, *Ezakiella* and *Enterococcus*. This composition was similar to that found by the Illumina platform, since these 10 most abundant genera were also among the most abundant genera detected by the Nanopore platform. In both cases, the top 10 genera represented more than 70% of the classified reads. However, relative abundance of each bacterium did not correlate between these two platforms, with intraindividual variations of up to 50 percentage points in some cases. Results suggest that the effect of the sequencing platform on the characterization of seminal microbiota is not very large at the phylum level, with slightly variances in *Firmicutes* and *Actinobacteria*, but presents differences at the genus level. These differences could alter the composition and diversity of bacterial profiles or posterior analyses. This indicates the importance of conducting multi-platform studies to better characterize seminal microbioma.

## 1. Introduction

The *16S rRNA* gene is a highly conserved gene found only in bacteria and present in some hypervariable regions that allow for differentiation of bacterial species based on sequence analysis. In recent years, *16S rRNA* gene sequencing studies using next-generation sequencing techniques (NGS) have shown that human seminal microbiota is mainly composed of four bacterial phyla: *Firmicutes*, *Bacteroidetes*, *Proteobacteria* and *Actinobacteria*. Although different authors have observed different relative abundances of these phyla, they agree that the most abundant group is *Firmicutes* [1,2]. The microbial composition at the genus level shows high interindividual variability, with different relative abundances for each taxon; however, the most observed taxa coincide in most cases [1,2,3,4,5,6]. Among the most frequently observed genera are *Lactobacillus*, *Corynebacterium*, *Acinetobacter*, *Prevotella*, *Enterococcus*, *Veillonella*, *Streptococcus*, *Porphyromonas*, *Staphylococcus*, *Finegoldia*, *Ralstonia* and *Pseudomonas*. Some components of these seminal microbiota have been associated with sperm quality parameters, suggesting a possible effect on male fertility [7,8,9,10,11,12,13,14,15,16,17,18,19]. The *Prevotella* genus, for instance, has been associated with poor seminal quality factors, principally with a reduction in sperm motility [8,15,17,18]. The mechanisms via which these bacteria interfere with these sperm quality parameters are still unknown but may be related to bacterial metabolism or enzymatic activities [11]. 

It has been established that methodological limitations affect the accuracy of microbiota studies [20,21,22,23,24], especially in cases of low biomass, such as seminal plasma. The chosen sequencing methodology, the targeted *16S rRNA* hypervariable region, the different analysis strategies and other variables may result in variations and contradictory results between research groups. In fact, the studies conducted to date diverge greatly in the amplified hypervariable region, and although most of them use the Illumina platform, not all of them use the same device for sequencing (Table 1). It is therefore of great interest to design full-sequence cross-platform studies to obtain contrasting and relevant information on the role of the microbiota on human male fertility.

Nanopore sequencing (ONT, Oxford Nanopore Technologies, Oxford, UK) is a third-generation sequencing technology that can generate long reads, such as the complete *16S rRNA* gene sequence. Recent comparative studies have shown that this technology has promising results in identifying the composition of the microbiota, which are comparable to those obtained with other already established NGS platforms such as Illumina or IonTorrent PGM, although Nanopore platform shows a tendency to find greater bacterial diversity [23,25,26].

In the present study, seminal microbiota composition of infertile idiopathic patients and donors was analyzed by sequencing the full-length *16S rRNA* gene using ONT’s MinION platform. The patients’ and controls’ seminal microbiota composition had been previously analyzed using the Illumina MiSeq platform [11]. We present a comparison of the results obtained with these two platforms.

## 2. Results

### 2.1. Seminal Microbiota Composition

The *16S rRNA* sequencing generated an average of 129,469.73 quality reads per sample, corresponding to 55.07% of the total reads generated by the platform. These reads had an average length of 1249.43 bp, and an average Nanopore quality score of 9.63 (encoded with ASCII characters from 33 to 126) [27], covering an average depth of 104,363.46 (LN/G, where L is the read length, N is the number of reads and G is the gene length [28]). Contamination controls from sample collection, DNA extraction and PCR amplification produced an average of 40.33 reads per sample, a substantially smaller number, leading to the assumption that the abundances found in the samples are not compromised; therefore, no correction was applied. Bacterial profiles observed in contamination controls are shown in Appendix A. The bacterial abundances resulting from the mock community analysis can be seen in Figure 1.

Taxonomic classification showed an average accuracy of 87.57% and only 2.84% of the reads remained unclassified. According to the taxonomic results, the seminal microbiota of the whole sample was mainly composed of four phyla: *Firmicutes* (~67%), *Proteobacteria* (~21%), *Bacteroidetes* (~5%) and *Actinobacteria* (~5%). Considering only taxa above 0.01% global relative abundance, 95 genera were identified in study samples, among which the most abundant were *Peptoniphilus*, *Finegoldia*, *Staphylococcus*, *Anaerococcus*, *Campylobacter*, *Prevotella*, *Streptococcus*, *Lactobacillus* and *Ezakiella* (Figure 2, Appendix A), all of them with >3% of global relative abundance. These nine genera accounted for 70.10% of all classified reads. However, differences were observed in the relative abundances of these taxa between individuals (Figure 2).

### 2.2. Seminal Microbiota Estructure

A clustering analysis of all samples allowed the identification of two bacterial profiles at each taxonomic level (Appendix A). At the phylum level, both profiles were predominantly composed of *Firmicutes*. However, phylum-profile 2 also presented relatively higher abundances of three other phylum: *Proteobacteria*, *Bacteroidetes* and *Tenericutes* (Figure 3A). This difference between profiles was also observed in alpha diversity, which was significantly higher in phylum-profile 2 (*p* = 0.011).

At the family level, family-profile 1 was dominated mainly by *Peptoniphilaceae* while family-profile 2 was composed of a lower abundance of *Peptoniphilaceae* compared to the previous profile, but a more prominent presence of *Staphylococcaceae*, *Prevotellaceae*, *Campylobacteraceae*, *Lactobacillaceae* and *Bacillaceae* (Appendix A). Alpha diversity was also found to be significantly different (*p* = 0.018).

### 2.3. Relative Abundance Comparison between Sequencing Platforms

At the genus level, genus-profile 1 showed a high abundance of *Peptoniphilus*, *Finegoldia* (both in similar presence), and *Anaerococcus*. Genus-profile 2 was more homogeneous in composition and included *Staphylococcus* (as the more predominant genus), *Prevotella*, *Streptococcus*, *Campylobacter*, *Lactobacillus*, *Finegoldia* and *Peptoniphilus* with similar abundance between them (Figure 3B). The alpha diversity of the two genus-level profiles showed no differences between them (*p* = 0.096).

In the comparison between the relative bacterial abundances detected by the Illumina sequencing platform and those detected by the Nanopore platform, only taxa with an overall relative abundance greater than 0.05% on at least one platform were used. A Wilcoxon test for paired samples was performed for phylum, family, and genus taxonomic levels in order to assess the similarity between the bacterial profiles of each individual detected by each sequencing platform (Appendix A).

At the phylum level, no significant differences in the relative abundances of *Bacteroidetes* and *Proteobacteria* were found between the two sequencing platforms. There were significant differences in the *Firmicutes* phylum (~59% in Illumina and ~67% in Nanopore) and very significant differences in the rest of the phyla, including *Actinobacteria* (~8% in Illumina and ~5% in Nanopore).

At lower levels, it was observed that the differences identified at the phylum level did not occur in all families and genera. In fact, many of the most abundant taxa in seminal plasma had similar abundances on both sequencing platforms and only a few taxa were responsible for the differences observed at the phylum level. Within *Bacteroidetes*, differences between platforms could be found in some taxa, such as the *Flavobacteriaceae* or *Bacteroidaceae* families and their main genera; however, these represent a small proportion of the total group. Thus, the largest family was *Prevotellaceae*, a group that showed no significant differences between Illumina and Nanopore results, even at the genus level. Although the phylum *Proteobacteria* showed no significant differences between the two sequencing platforms, some of its lower taxa did. The relative abundances of *Enterobacteriaceae* or *Moraxellaceae*, for instance, were significantly different between the two platforms (Appendix A), the last mainly because of the differences observed for the *Acinetobacter* genus. Other families, such as *Bradyrhizobiaceae* or *Neisseriaceae* also presented differences; however, their weight within the phylum was lower (<0.5%). On the other hand, the most representative taxa such as the families *Campylobacteraceae* (and its genus *Campylobacter*), *Burkholderiaceae* (mainly *Cupriavidus* and *Ralstonia* genera) or the genus *Moraxella* showed no differences between the relative abundances detected in Illumina and in Nanopore platforms.

The differences observed in the phylum *Firmicutes* were also observed in some of its most representative taxa. The *Staphylococcus*, *Lactobacillaceae* and *Peptostreptococcaceae* families, and their genera *Staphylococcus*, *Lactobacillus* and *Filifactor*, respectively, were detected with significantly different relative abundances between the two sequencing platforms (Appendix A). Other less abundant genera, such as *Megasphera* or *Murdochiella*, also showed differences. However, the largest family, *Peptoniphilaceae*, together with its most abundant genera in the seminal plasma (*Finegoldia*, *Peptoniphilus* and *Anaerococcus*), did not show differences between the Illumina and Nanopore platforms, nor did other relevant families such as *Streptococcaceae* or *Veillonellaceae*. Finally, the *Corynebacterium*, *Actinotignum*, *Mobiluncus*, *Schaalia* and *Gardnerella* genera were the main taxa where significant differences were found within the phylum *Actinobacteria*; however, at family level, they were not observed.

## 3. Discussion

The interest in the characterization of seminal plasma microbiota has recently grown due to its potential implications on dysbiosis in male fertility. Correlations have been observed between some bacterial genera and the most relevant seminal and sperm parameters in the diagnosis of male infertility, including sperm motility and concentration, DNA damage or oxidative stress [1,3,5,18]. Detailed descriptions of the composition of this microbiome have been possible thanks to the development of next-generation sequencing techniques; however, there is still a lack of consensus regarding the abundance of bacteria, and about which bacteria are most abundant in semen. Methodological differences in the studies carried out to date may explain the differences reported by different authors (Table 1). In this study, a multi-platform analysis using the full-length *16S rRNA* gene sequencing was performed to observe the impact of the sequencing platform on the composition of the microbiota observed. The sequencing platforms used were Illumina MiSeq and ONT MinION. The same 56 samples were divided before the library preparation and characterized with both platforms, ensuring comparable results and that any differences were solely due to the sequencing technique. Results from the Illumina MiSeq platform have been published previously [11].

Characterization using the ONT MinION platform revealed that the seminal microbiota is composed of four bacterial phyla: *Firmicutes*, *Proteobacteria*, *Actinobacteria* and *Bacteroidetes*. The proportions observed for each phylum are similar to those observed using the Illumina platform [11], although MinION has detected a significantly higher amount of *Firmicutes* and a lower amount of *Actinobacteria* (Figure 4 and Appendix A). These results are consistent with most previous studies [10,12,13,14,15,17,18,19]. The analysis of the mock community showed an underrepresentation of the genera belonging to the phylum *Proteobacteria* by both platforms; therefore, the abundance of this phylum could be higher than that observed in this study.

Illumina’s analysis identified 804 different bacterial genera, while ONT was able to identify 963. A total of 397 of them were found in both platforms. However, only 168 of 804 Illumina genera (20.90% of the total) contained at least 0.01% of the total reads, and ONT had even fewer genera above the 0.01% abundance threshold with 97 out of 963 (10.07%). Of the genera above the abundance threshold, 65 were found in both platforms (Figure 5). Genera with lower frequency may be sequencing or classification artifacts due to error rates of the platforms used and taxonomic classification algorithms. It is known that the ONT platform produces small errors in the reads that are then misclassified as another genus in post-informatics analysis [27,29]. In both platforms, around 90% of the classified reads correspond to approximately the top twenty most abundant genera.

The phylum *Firmicutes* is the most abundant in seminal fluid [1]. In our samples, regardless of the sequencing method used, the main representatives of this phylum were bacteria from the family *Peptoniphilaceae*: *Peptoniphilus*, *Finegoldia* and *Anaerococcus*, three of the four genera with the highest relative abundance (Appendix A). These genera have not been previously identified as the most abundant in the seminal microbiota; however, several authors described them as recurring bacteria [7,8,10,12,13,15]. The Wilcoxon test shows that there are differences in the relative abundances of *Staphylococcus* between the two sequencing platforms (Appendix A and Figure 6) observing a greater abundance in the analysis by the Nanopore platform (also observable at the level of its family *Staphylococcaceae*), where it was the third most abundant genus. The mock community analysis showed that the Nanopore sequencing methodology produced an overrepresentation of *Staphylococcus*, while the one used by Illumina produced an underrepresentation (Figure 1). This could explain the differences that were found in seminal samples. Other authors have already described this genus as one of the most abundant in seminal fluid, even noting it as the most abundant in some cases [7,9,10,13,15]. However, the overrepresentation of *Staphylococcus* by ONT MinION has been described recently in skin samples [30]; therefore, it could be an artifact of the platform. 

Another taxon of *Firmicutes* phylum that had a different abundance in the Nanopore platform is *Lactobacillus* (and its family *Lactobacillaceae*) (Appendix A and Figure 6). Again, this bacterial group is one of the most described in the literature and several authors have detected it as the most abundant in seminal fluid [7,8,12,14,15,16,17,19]. Despite being detected in greater abundance than the results with Illumina sequencing, the relative abundance levels observed in this cohort remain lower than those observed by most authors and are similar only to those observed by Monteiro et al. and Lundy et al. [13,18]. The mock community analysis showed that both sequencing platforms displayed lower than expected abundances of this genus and that *Lactobacillus* relative abundance was better detected by the Illumina platform. Finally, the *Peptostreptococcaceae* family also presents significant differences in abundance between the two platforms, probably due to the differences in its most abundant genus in the seminal fluid, *Filifactor* (Appendix A). Although there are more *Firmicutes* genera and families with differences between sequencing platforms, most of the taxa with higher abundance are detected similarly by both Illumina and Nanopore. Therefore, the differences observed in the *Firmicutes* phylum are likely explainable by differences in the detection of the abundance of *Staphylococcus*, *Lactobacillus* and *Peptostreptococcaceae*, which are the most abundant taxa among those that present differences. In addition, the lower the abundance of a taxon, the more likely it is that the observed differences come from errors in the post-sequencing bioinformatics analysis rather than from the sequencing technique used; therefore, bacteria with low abundances should be analyzed with caution in these studies. It should be noted that ONT MinION has also been able to detect the genus *Ezakiella*, described for the first time in seminal samples by our group in an earlier study using the Illumina platform [11].

In our samples, the most abundant bacterial genus in *Actinobacteria* was *Corynebacterium*, a frequently described taxon in seminal fluid [1], which had significantly different abundances using the Nanopore and Illumina platforms (Appendix A and Figure 6). In fact, most of the bacteria in this phylum presented differences between sequencing platforms: *Actinotignum*, *Mobiluncus* and *Schaalia* from the *Actinomycetaceae* family; *Gardnerella* of the *Bifidobacteriaceae* family; and other families such as *Intrasporangiaceae*, *Micrococcaceae* and *Promicromonosporaceae*; most of them previously described [7,8,9,10,12,13,14,15,16,17,18,19].

Although *Proteobacteria* and *Bacteroidetes* abundances showed no differences between Illumina and Nanopore at the phylum level, differences were detected in some of their families or genera (Appendix A). In this cohort, *Proteobacteria* were especially abundant, with a notable presence of *Campylobacter*, *Ralstonia* and *Moraxella*. Of these, the *Campylobacter* genus was the most abundant, and is the most frequently described in the literature [8,12,14,15]. Both *Campylobacter* and *Ralstonia* had similar abundances using both Nanopore and Illumina platforms (Figure 6). Significant cross-platform differences have been observed between the abundances detected by Nanopore and Illumina in the family *Moraxellaceae* and its genus *Acinetobacter*, but not in its genus *Moraxella*. *Moraxella* was first described in seminal fluid by Monteiro [13] and its possible beneficial relationship with male fertility was described in the Illumina platform analysis carried out by our group [11]. Differences have also been found for other taxa of the phylum *Proteobacteria* including the genera *Oligotropha* and *Rhodopseudomonas* (as well as their family *Bradyrhizobiaceae*), *Neisseria* (and its family *Neisseriaceae*), *Steroidobacter* (and *Steroidobacteraceae*), and the families *Sphingomonadaceae*, *Oxalobacteraceae* and *Enterobacteriaceae*. However, the impact of these differences is not enough to be observed at the phylum level, probably due to the low presence of most of these taxa. Finally, in seminal plasma, the phylum *Bacteroidetes* is mainly composed of the genus *Prevotella*, one of the most widely described bacteria in this environment and one of the most important candidates for a possible relationship with male fertility [7,8,9,10,12,13,14,15,16,17,18,19]. In accordance with the observation at the phylum level, no differences were observed between the abundances detected by the two sequencing platforms (Figure 6). Differences were observed in other taxa with low representation such as *Bacteroides* or *Flavobacterium*.

In summary, the majority of the most abundant taxa in seminal plasma show no difference whether analyzed with the Illumina MiSeq platform or with the ONT MinION platform (Figure 6), providing evidence that third-generation Nanopore technology is equivalent to the technologies already utilized in this area. The abundance of each bacterial group per individual was detected in similar proportions with the exception of some specific taxa: the genera *Staphylococcus*, *Lactobacillus*, *Corynebacterium* and the families *Peptostreptococcaceae* and *Moraxellaceae* (mainly *Acinetobacter*), which had different proportions, depending on the platform. Abundance differences between the two sequencing platforms are also observed in taxa with lower biomass, although these bacteria will contribute a very low percentage of reads in the total count. With so few copies of the *16S rRNA* gene, any errors in sequencing or taxonomic classification or the filtering of reads for low quality can cause their relative abundance to vary substantially. Furthermore, some of these low-abundance bacteria may be classification artifacts of more abundant species.

Regarding bacterial structure, two profiles were observed for each taxonomic level. These two profiles had also been observed in the Illumina study [19]. Phylum-level profiles had a very similar composition to the observed by Illumina. However, family- and genus-level profiles showed some differences between platforms. Family profiles coincided in the predominance of *Peptoniphilaceae* above other taxa; however, the composition of the other taxa differed between Nanopore and Illumina. The profile differences accentuated at the genus level, (Appendix A and [19]) and seemed to be greater at lower taxonomic levels. Thus, and as we have just discussed, the composition of the bacterial profiles could be altered by changes in the abundance of some taxa.

The origin of differences observed in these bacterial groups is not known, but may be due to the methodological differences of each technique. Illumina, for example, performs sequencing-by-synthesis and this approach is associated with intrinsic errors of the technique such as color interference, phasing or dimming [31]. During sequencing-by-synthesis, fluorochromes are used to indicate the nucleotide that is incorporated in each round of synthesis; however, if there is interference with adjacent sequences or clusters, nucleotide exchange errors can be incorporated into the sequence. Phase errors occur when there are problems with the terminators of the incorporated nucleotides, which can cause two nucleotides to be incorporated in a row or nucleotides not to be incorporated in a round of synthesis, resulting in small insertions or deletions. If the terminator of a new nucleotide is completely immovable, dimming of this cluster can occur, leading to an incomplete or error-prone sequence. Nanopore technology produces biases with a similar effect, and with a higher frequency than those produced by Illumina [27]. Although all NGS technologies exhibit GC-bias, Nanopore seems to present more problems in GC-rich sequences. Low-GC content species tend to produce better quality sequences, and consequently, GC-rich species will have reads discarded for poor quality in higher proportions. In addition, high GC contents also affect the sequencing of homopolymers. The Nanopore basecaller software, in charge of translating the electrical signals of the pores into nucleic sequences, although optimized nowadays, still presents errors that lead to insertions, deletions, or mismatches. Any of these small changes introduced by the two sequencing platforms, if they occur in key regions of the *16S rRNA* sequence used to classify it into one bacterial group or another, can lead to classification errors that slightly alter the abundances of bacteria.

The differential abundances observed in some bacterial genera and families could bias future analyses used to define a possible relationship between seminal microbiota and male fertility. In fact, this could be one of the causes of the lack of consensus between different authors regarding the effect of bacteria on fertility parameters, since analysis of seminal microbiota has not yet been standardized and there is disparity in the techniques used. More full-sequenced multi-platform studies are necessary in order to discern if the differences observed within the same seminal sample, analyzed by different platforms, are substantial enough to influence our interpretation of the effect of bacterial taxa on seminal quality.

## 4. Materials and Methods

### 4.1. Sample Collection

Seminal samples were collected from 56 subjects: 42 idiopathic normozoospermic infertile patients with no apparent female factor as a possible cause of the couple’s infertility, and 14 control samples from semen donors (further details of this cohort are available at Appendix A and Garcia-Segura et al., 2022 [11]). Samples were obtained after 2–5 days of sexual abstinence from patients attending the *Instituto de Fertilidad* of Palma (Mallorca, Spain) or *Universitat Autònoma de Barcelona* (UAB, Bellaterra, Spain), where fertility status was assessed. A specific previously described protocol was used to prevent bacterial contamination and preserve the samples [11]. A mock community (ZymoBIOMICS Microbial Community DNA Standard; Zymo Research, Irvine, CA, USA) was employed as a positive control. Informed consent was obtained from all donors, and the study was approved by the Parc Taulí Hospital ethics committee, with registration number 2014676, according to the Declaration of Helsinki.

### 4.2. Microbiome DNA Extraction

Bacterial DNA extraction from seminal samples and mock community was performed with ZymoBIOMICS DNA Microprep kit (Zymo Research, Irvine, CA, USA). To allow the release of DNA, a bead beating in lysis solution was performed to lyse cell walls and membranes. Then, slight modifications were included into the manufacturer’s protocol, to filter samples. All supernatant was filtered in Zymo-Spin III-F by centrifugation at 8000× *g* for 1 min and washing repeatedly in a Zymo-Spin IC-Z column to purify DNA before elution. Sterile gloves and a horizontal laminar flow cabinet, previously sterilized with DNA-degrading products, and UV irradiation was used during extraction step. A sterile swab was placed inside the cabinet as a negative environmental control and a blank control was included to observe possible *kitoma* contamination.

### 4.3. 16S rRNA Gene Sequencing

Full-length *16S rRNA* gene (V1-V9 hypervariable regions) was amplified via two consecutive PCRs to characterize seminal microbiota. First amplification was performed with modified 27F and 1492R universal *16S rRNA* primers, which include a specific tag (27F: 5′-TTTCTGTTGGTGCTGATATTGCAGRGTTTGATYHTGGCTCAG; 1492R: 5′-ACTTGCCTGTCGCTCTATCTTCTACCTTGTTAYGACTT, tag underlined) allowing subsequent indexing. The PCR Mix contained 10.3 µL nuclease-free water, 4 µL 5× buffer, 2 µL 2 mM dNTP, 0.8 µL 10 µM forward primer, 1.6 µL 10 µM reverse primer, 0.3 µL 2 U/µL Phusion Hot Start II Taq HIFI polymerase (Thermo Fisher Scientific; Waltham, MA, USA) and 1 µL DNA template per sample. The thermocycler was set with an initial denaturation at 98 °C for 30 s, followed by 25 cycles of denaturation at 98 °C for 15 s, annealing at 62.5 °C for 15 s and extension at 72 °C for 45 s. The program was completed with a final extension at 72 °C for 7 min. Blank control was included to observe possible *kitoma* contamination.

In order for the results obtained by the two sequencing platforms (Illumina and ONT) to be comparable, an identical protocol was established for the extraction and first amplification of the samples, to minimize the possible alterations produced by the methodological procedure. Then, two aliquots of each sample were made to be sequenced by two different platforms: Illumina MiSeq and ONT MinION. The protocol for Illumina sequencing is described in Garcia-Segura et al., 2022 [11], while the protocol for ONT sequencing is described below.

DNA indexing was performed by a second PCR with the PCR Barcoding kit (Ref. SQK-PBK004; ONT, UK), which targets a specific tag incorporated in the first amplification and includes the Nanopore sequencing adapters. The Nanopore Community’s Four-primer PCR protocol (SQK-PSK004/SQK-PBK004—FFP_9038_v108_revG_27Jun2017) and SequalPrep polymerase (Thermo Fisher Scientific) were used for the indexing. The PCR Mix contained 4.67 µL nuclease-free water, 1 µL 10× buffer, 0.55 µL 5.5% DMSO, 1 µL 10× Enhancer, 0.1 µL 50 mM MgCl_2_, 1.5 µL 10 µM primer mix, 0.18 µL 5 U/mL SequalPrep polymerase and 1 µL of a 1:10 dilution of the first PCR’s DNA product. The thermocycler was set with an initial denaturation at 94 °C for 60 s, followed by 20 cycles of denaturation at 94 °C for 30 s, annealing at 62 °C for 30 s and extension at 65 °C for 75 s. The program ended with a final extension at 65 °C for 5 min.

A purification step with Agencourt AMPure XP beads (Beckman Coulter, Brea, CA, USA) was applied to eliminate non-specific products derived from the PCR Barcoding kit. Magnetic bead solution was added into the PCR product solution in a 1:2 proportion, incubated with agitation for 5 min at room temperature and briefly centrifuged. Using MagnaRack Magnetic Separation Rack (Thermo Fisher Scientific), amplification products were separated magnetically, supernatant was removed and DNA was washed twice with 200 µL of 70% ethanol. DNA was dried for 2 min to evaporate residual ethanol and was suspended in 10 µL 10 mM Tris-HCl, supplemented with 50 nM NaCl at pH 8.0 to separate beads from DNA. After 5 min incubation at room temperature, beads were removed using MagnaRack and DNA was diluted to a final concentration of 4 nM.

Continuing the Four-primer PCR protocol, the samples were prepared to be sequenced using PCR Barcoding kit adapters. Rapid 1D sequencing adapters (RAP) were incorporated into the DNA library at a 1:10 proportion and incubated for 5 min at room temperature. After that, samples were kept on ice (2–8 °C) until sequencing. Pooled libraries were sequenced on the MinION system (ONT, UK) using R9.4.1 flowcells (ONT, UK), following the manufacturer’s instructions.

### 4.4. Bioinformatics Analysis

Taxonomic classification of bacterial reads was performed using EPI2ME 16S Workflow (v2022.01.07) (EPI2ME, ONT), which uses the NCBI taxonomic database, with the following settings: 1400–1700 bp read length range, minimum coverage of 30%, minimum identity of 77% and with a maximum of 3 target sequences for BLAST alignment. The quality control of sequencing data was evaluated by the taxonomic classification software itself. Raw relative abundances at phyla, families and genera taxonomic levels were formatted using an in-house script in PERL (https://www.perl.org/). Dataset obtained from sequencing and associated metadata, is available on-line in the “Dipòsit Digital de Documents (DDD), Universitat Autònoma de Barcelona” (https://doi.org/10.34810/data680 (accessed on 15 March 2023)).

### 4.5. Statistical Analysis

Statistical analyses were conducted using the IBM SPSS Statistics 26.0 software (IBM Corp., Armonk, NY, USA), and GraphPad Prism v.8 (GraphPad Software, La Jolla, CA, USA) and the level of significance was set at *p* < 0.05. Statistical normality of bacterial abundances was checked using Shapiro–Wilk test. Comparisons of the relative abundance for phylum, family, and genus taxonomic levels between sequencing platforms were performed with the Wilcoxon test for paired samples, considering those taxa with global relative abundances above 0.05% on at least one platform.

To investigate the presence of distinct microbiologic profiles at the phyla, families or genera taxonomic levels, a cluster analysis was conducted considering the relative abundances of the bacteria identified with the ONT technology in the whole sample, following the enterotyping tutorial in R (EMBL3) [32,33] with the R environment. Briefly, the between-groups linkage method based on the Euclidean distance of 56 samples was calculated considering the composition data for phylum, family and genus taxonomic levels (vegan package, version 1.6-0). Partitioning around medoids (PAM) clustering was performed based on the obtained distance matrix (cluster package, version 2.1.4). The optimal number of clusters was chosen by maximizing the Calinsky–Harabasz index [34], and the obtained cluster was validated via prediction strength [35] and silhouette index [36] (clusterSim package, version 0.50-1; cluster package, version 2.1.4). Clustering analysis was performed regardless of fertility status of samples. 

## 5. Conclusions

Full-length *16S rRNA* gene sequencing using the ONT MinION platform is a valid methodology for the characterization of seminal microbiota, with similar results to Illumina MiSeq, but with some differences to consider. The multi-platform study has provided further evidence on the composition of the seminal plasma microbiota at the phylum level, which is composed of *Firmicutes*, *Proteobacteria*, *Actinobacteria* and *Bacteroidetes*. In the present study, the use of the ONT MinION has enabled us to identify the most abundant bacterial genera as *Peptoniphilus*, *Finegoldia*, *Staphylococcus*, *Anaerococcus*, *Campylobacter*, *Prevotella*, *Streptococcus*, *Lactobacillus* and *Ezakiella*, which largely coincide with those observed using Illumina MiSeq in the same cohort. 

However, differences in the relative abundances of some major bacterial groups have been observed between the two sequencing platforms: the genera *Staphylococcus*, *Lactobacillus* and *Corynebacterium* and the families *Peptostreptococcaceae* and *Moraxellaceae*. It is currently unknown whether these changes may alter the interpretation of the effect of bacterial taxa on seminal quality; therefore, more full-sequence multi-platform studies are needed to avoid biases.

## Figures and Tables

**Figure 1 ijms-24-07867-f001:**
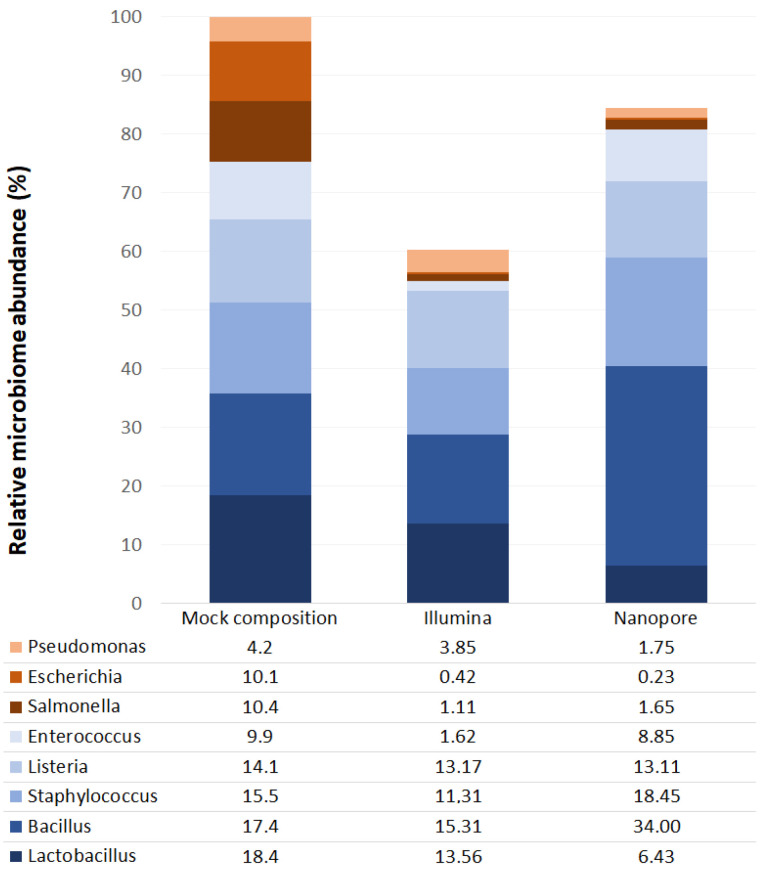
Relative abundances of bacteria identified in mock community samples from Illumina MiSeq (Illumina) and ONT MinION (Nanopore) platforms. The X-axis shows the theoretical mock composition based on the “16S Only” supplier’s specifications (**left**), as well as the sequencing results of the mock community using the Illumina platform (**middle**) and the Nanopore platform (**right**). The Y-axis corresponds to the relative abundance of each taxon in percentage.

**Figure 2 ijms-24-07867-f002:**
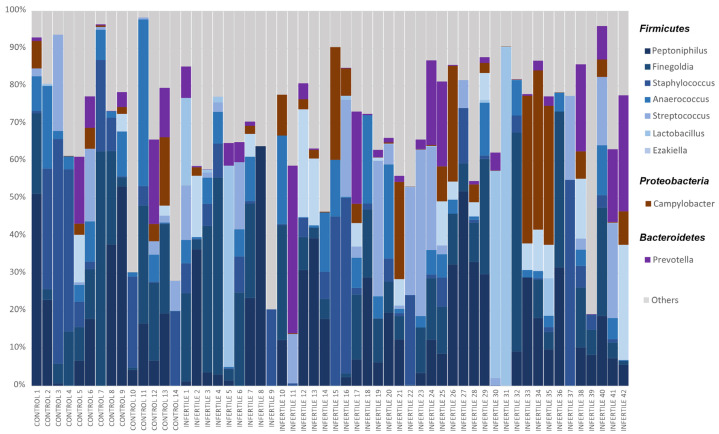
Relative abundances of bacteria in seminal microbiome obtained by Nanopore sequencing from control donors and idiopathic infertile patients at the genus level. The X-axis shows each individual of the cohort, and the Y-axis corresponds to the relative abundance of each taxon in percentage. Only bacteria with a relative abundance of over 3% in the whole sample were included.

**Figure 3 ijms-24-07867-f003:**
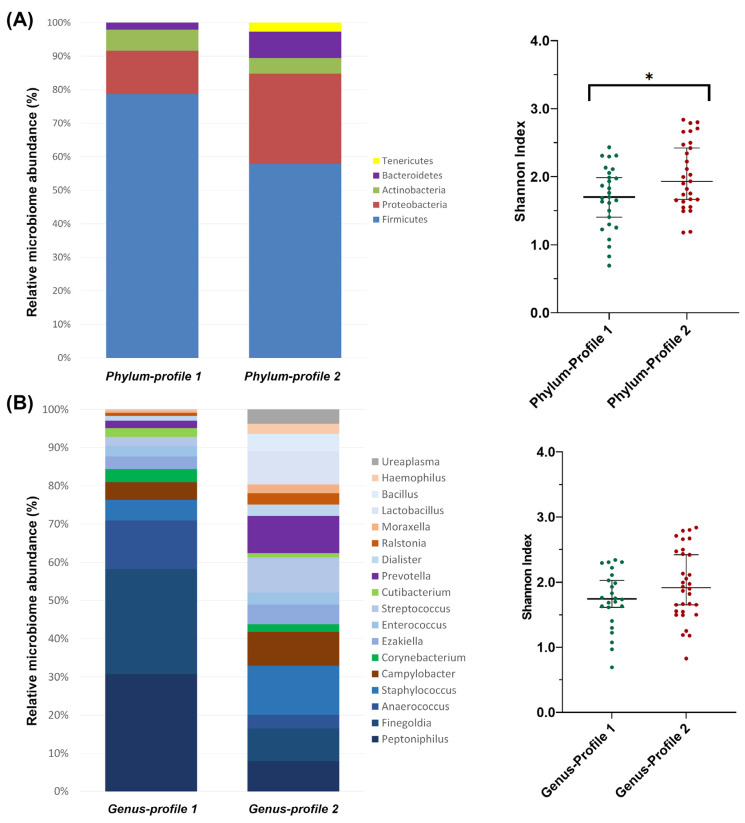
Microbiome profiles’ composition from clustering analysis of (**A**) phylum and (**B**) genus taxonomic levels. The bar-plots show the relative abundance in percentage of the most representative bacteria of each profile. The scatter dot-plots display the alpha diversity distribution of each profile, where the Y-axis represents the Shannon index. Thin horizontal lines delimit the 95% confidence interval (CI), whereas the thick horizontal mark denotes the median value. Asterisks indicate statistically significant differences between platforms according to a unpaired *t*-test.

**Figure 4 ijms-24-07867-f004:**
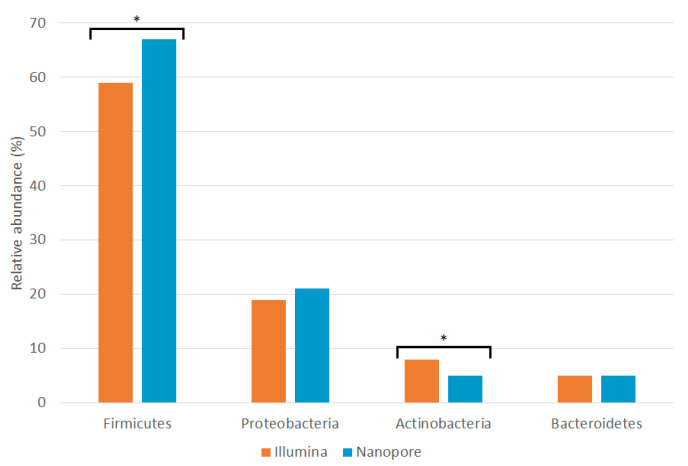
Relative abundances of seminal phyla detected by Illumina MiSeq (orange; [11]) and ONT MinION (blue; this study) sequencing platforms. Asterisks indicate statistically significant differences between platforms according to a Wilcoxon signed-rank test (Appendix A).

**Figure 5 ijms-24-07867-f005:**
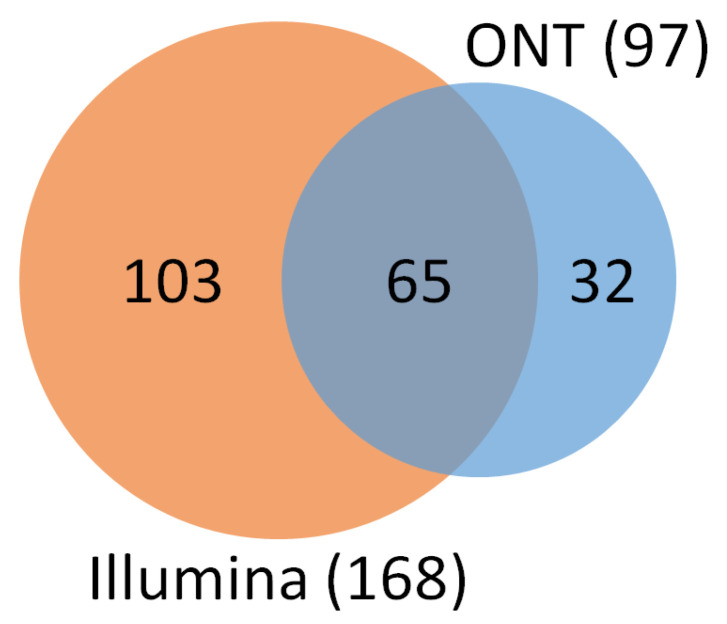
Number of identified genera by Illumina MiSeq (orange) and ONT MinION (blue) sequencing platforms with at least 0.01% of the total relative abundance. The two platforms had 65 genera in common.

**Figure 6 ijms-24-07867-f006:**
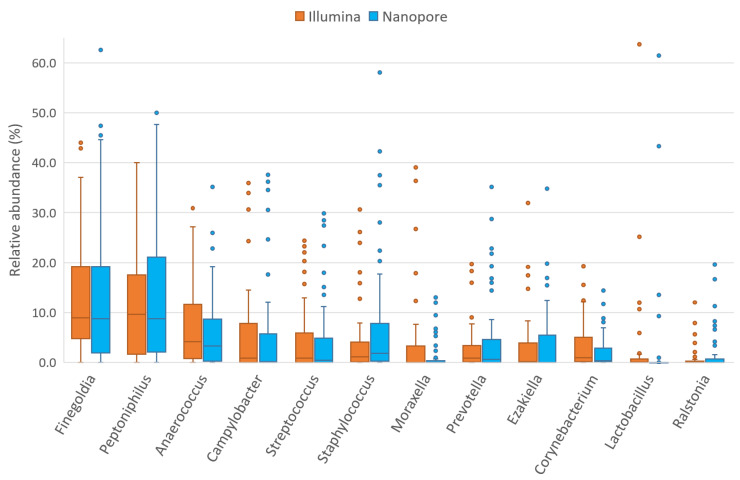
Relative abundances of 12 most abundant seminal genera detected by Illumina MiSeq (orange; [11]) and ONT MinION (blue; this study) sequencing platforms.

**Table 1 ijms-24-07867-t001:** Methodologies used in characterization of seminal microbiota. Description of the amplified *16S rRNA* gene hypervariable regions and the sequencing platform used are shown for each case.

Study	Amplified *16S* Region	Sequencing Platform
Hou et al., 2013 [7]	V1–V2	Roche 454 pyrosequencing
Weng et al., 2014 [8]	V4	Illumina Miseq
Mändar et al., 2017 [12]	V6	Illumina HiSeq2000
Monteiro et al., 2018 [13]	V3–V6	Ion Torrent PGM 316
Chen et al., 2018 [14]	V4	Illumina HiSeq 2000
Baud et al., 2019 [15]	V1–V2	Illumina MiSeq
Amato et al., 2020 [16]	V3–V4	Illumina MiSeq
Yang et al., 2020 [17]	V1–V2	Illumina HiSeq 2500
Lundy et al., 2021 [18]	V3–V4 *Shotgun*	Illumina MiSeq Illumina NovaSeq 6000
Okwelogu et al., 2021 [19]	V4	Illumina NextSeq 500
Bukharin et al., 2022 [9]	Not specified	Illumina MiSeq
Yao et al., 2022 [10]	V3–V4	Illumina MiSeq
Garcia-Segura et al., 2022 [11]	V1–V9	Illumina MiSeq

## Data Availability

Dataset obtained from sequencing and associated metadata are available on-line (https://doi.org/10.34810/data680) in the “Dipòsit Digital de Documents (DDD), Universitat Autònoma de Barcelona”.

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
