# Peer review of "Characterization of Seminal Microbiome of Infertile Idiopathic Patients Using Third-Generation Sequencing Platform"

_ijms, 2023, doi:10.3390/ijms24097867_

Round 1

Reviewer 1 Report

This article, titled Characterization of seminal microbiome of infertile idiopathic patients using third-generation sequencing platform, compares the sequencing results of the semen microbiome of infertile patients and control patients between two deep sequencing platforms: Illumina MySeq and MinION. There is a difference between both platforms at the genus level, within which there are differences in composition and bacterial diversity. This underscores the importance of using different sequencing platforms when characterizing microbiomes. 

Minor concerns: 

1) The simulation differs from the theoretical composition based on genomic DNA: Listeria monocytogenes - 12%, Pseudomonas aeruginosa - 12%, Bacillus subtilis - 12%, Escherichia coli - 12%, Salmonella enterica - 12%, Lactobacillus fermentum - 12 %, Enterococcus faecalis - 12%, Staphylococcus aureus - 12%, Saccharomyces cerevisiae - 2% and Cryptococcus neoformans - 2% according to the supplier. Why does the composition of the mock shown in Fig. 1 change?

 2) What sequencing platform was used for the mock analysis? 

3) Fig. 2 shows the relative abundance of sequencings obtained with Nanopore? I propose to indicate the origin of the data in the footer of the figure. 

4) The relative abundance of the bacteria compared to  11reference is very different in Illumina-Myseq sequencing. Why were Actinobacteria and Moraxella not included or considered elsewhere? It is suggested to add actinobacteria as they will be included in later analyzes (Figs. 3 and 4). 

5) Proteobacteria and Bacteroidetes are absent in 7/14 control samples and 11/42 patient samples, is this common in semen samples? 

6) Was the clustering analysis performed with the Illumina or Nanopore sequences? 

Reviewer 2 Report

In this authors' study, a multiplatform analysis using full-length 16S rRNA gene sequencing was performed to observe the impact of sequencing platform on observed microbiota composition.

Although the work is technically structured and uses ultramodern methods, it seems to escape the main topic - the effect of the sperm microbiota component on the quality and fertility of ejaculates.\

I believe that complementary information should be provided in this regard.

For the accuracy and substantiation of the need for the study, I believe that more introductory information should be provided regarding the mechanisms by which bacteria interfere with sperm quality.

There are also spelling and technical editing mistakes in the text.

Reviewer 3 Report

The manuscript titled with “Characterization of seminal microbiome of infertile idiopathic 2 patients using third-generation sequencing platform” has compared the seminal microbiome sequencing results from both NGS and MinION. Authors found some differences in genera level from both sequencing platforms but not in phyla level. However, the manuscript is too descriptive and surficial. There are not much biological questions addressed in this manuscript.

1. what are the ages, weight and other factors from the patients or control group? Those factors affect the semen quality very much. Authors should list related parameters from the sample and control groups.

2. Authors already mentioned that amplification or DNA extraction will affect sequencing results. Why did authors not compare PCR conditions since you have used different second PCR conditions for both sequencing platforms?  

3. For the section 2.3, it would be better to have some graphs or tables to give a summarized comparison results from both sequencing platforms.

4. why do two sequencing platforms give different results in genera level? And how much can we trust the sequencing results if there is only one sequencing platform used?

Reviewer 4 Report

This manuscript provides a comprehensive study of seminal microbiome from controls and infertile idiopathic patients amplifying full-length sequencing of the 16S rRNA gene. For sequencing, the authors used MinION platform from Oxford Nanopore and Illumina's MiSeq sequencing platform. The results highlight the importance of conduct studies with different platforms due to the differences found in the microbial genus level. The manuscript is clearly written, figures are adequate, and references are relevant. The work is well presented and written in a logical way. I would like to make some suggestions to help improve the manuscript.

Introduction:

·         I would like to recommend that authors incorporate at the beginning of the introduction a couple of sentences indicating what 16S rRNS (16S ribosomal RNA) are and their importance in the identification of bacteria.

Results:

·         Would it be possible to perform a statistical analysis to analyze whether there are significant differences between the seminal microbiota of control individuals versus patients with idiopathic infertility?

·         I have reviewed the scientific article by Garcia-Segura et al., 2022 to find out the seminogram data of the control samples and patients with idiopathic infertility but I have not found this information. It would be interesting to show the seminogram data separately based on the control samples and the patients with infertility.

Materials and Methods:

·         Please, add information about how centrifugation is performed to separate sperm from seminal fluid.

Round 2

Reviewer 3 Report

The manuscript is improved a lot. I appreciated. 

Please check the legend inside of the figures, the font is too small to read, especially for figure 2 and 3.